# Unravelling the Effect of Provitamin A Enrichment on Agronomic Performance of Tropical Maize Hybrids

**DOI:** 10.3390/plants10081580

**Published:** 2021-07-31

**Authors:** Abebe Menkir, Ibnou Dieng, Wende Mengesha, Silvestro Meseka, Bussie Maziya-Dixon, Oladeji Emmanuel Alamu, Bunmi Bossey, Oyekunle Muhyideen, Manfred Ewool, Mmadou Mory Coulibaly

**Affiliations:** 1International Institute of Tropical Agriculture, Oyo Road, Ibadan PMP 5320, Nigeria; I.Dieng@cgiar.org (I.D.); W.Mengesha@cgiar.org (W.M.); S.Meseka@cgiar.org (S.M.); B.Maziya-Dixon@cgiar.org (B.M.-D.); O.ALAMU@CGIAR.ORG (O.E.A.); B.BOSSEY@CGIAR.ORG (B.B.); 2Institute for Agricultural Research, Ahmadu Bello University, Zaria PMB 1044, Nigeria; oyemuhyideen@yahoo.com; 3Crop Research Institute, Kumasi P.O. Box 3789, Ghana; manfredbewool@gmail.com; 4Institute de Economic Rurale, Bamako BP 258, Mali; madoumory@yahoo.fr

**Keywords:** provitamin A carotenoids, agronomic performance, stability, adaptability, trade-off

## Abstract

Maize is consumed in different traditional diets as a source of macro- and micro-nutrients across Africa. Significant investment has thus been made to develop maize with high provitamin A content to complement other interventions for alleviating vitamin A deficiencies. The current breeding focus on increasing β-carotene levels to develop biofortified maize may affect the synthesis of other beneficial carotenoids. The changes in carotenoid profiles, which are commonly affected by environmental factors, may also lead to a trade-off with agronomic performance. The present study was therefore conducted to evaluate provitamin A biofortified maize hybrids across diverse field environments. The results showed that the difference in accumulating provitamin A and other beneficial carotenoids across variable growing environments was mainly regulated by the genetic backgrounds of the hybrids. Many hybrids, accumulating more than 10 µg/g of provitamin A, produced higher grain yields (>3600 kg/ha) than the orange commercial maize hybrid (3051 kg/ha). These hybrids were also competitive, compared to the orange commercial maize hybrid, in accumulating lutein and zeaxanthins. Our study showed that breeding for enhanced provitamin A content had no adverse effect on grain yield in the biofortified hybrids evaluated in the regional trials. Furthermore, the results highlighted the possibility of developing broadly adapted hybrids containing high levels of beneficial carotenoids for commercialization in areas with variable maize growing conditions in Africa.

## 1. Introduction

Maize (*Zea mays* L.) accounts for 40% of the total cultivated cereal production area in Sub-Saharan Africa with more than 66% of the harvested grain used for human consumption [1,2]. In the last 10 years, the total harvested maize area in SSA has increased by nearly 60% [3]. Maize is consumed in a variety of local food products across regions in Africa, supplying 38% of the food calories to consumers [4]. The demand for maize in SSA is increasing because of the rapid population growth, urbanization, and the growing need for poultry feed [5]. Maize intake across Africa varies from 30 to more than 330 g/person/day [6], providing starch, protein, fat, micronutrients, including minerals and vitamins, fiber, and many phytochemicals with known health benefits [7]. However, the over dependence of millions of Africans on white maize-based diets that do not supply adequate amounts of these nutrients, including vitamin A, can adversely affect the health and well-being of individuals throughout their life span [8]. Yellow maize naturally accumulates some levels of provitamin A carotenoids in grains [9], but the concentrations of these carotenoids in widely grown tropical maize cultivars is too low to meet the daily requirements of consumers. Enhancing the native nutrient levels of food crops, including yellow maize, has therefore been advocated as a complementary approach together with the conventional interventions of supplementation, fortification, and dietary diversity to alleviate vitamin A deficiencies in developing countries [10]. 

Significant investments have been made to enrich maize with provitamin A through conventional breeding to reduce the incidence of vitamin A deficiency [10,11,12,13]. Provitamin A biofortified maize could also increase competitiveness through increased use in diverse food products, targeting niche markets, and the branding of products. The current breeding strategies to develop maize varieties and hybrids with high provitamin A content has primarily focused on exploitation of favorable alleles derived from diverse sources to promote the accumulation of higher levels of β-carotene at the expense of the synthesis of other carotenoids [11,13]. Using this approach, breeders have generated many tropical and sub-tropical maize inbred lines that meet or surpass the current breeding target of 15 μg g^−1^ and used them for developing hybrids with high concentrations of provitamin A [14,15]. However, the changes in accumulating other beneficial carotenoids to human health, including lutein and zeaxanthins, that occurred in provitamin A biofortified hybrids are rarely reported. 

Carotenoids are not only the main sources of provitamin A in the diet, but also play a prime functional role as accessory pigments in photosynthesis, increasing light capture and correcting the assembly of photosystems [16]. The xanthophyll cycle mediates increased tolerance to high light intensity and heat stress and protects the plant from photo-oxidative damage [17,18]. In addition, carotenoids control the synthesis of phytohormones to respond to biotic and abiotic stresses and regulate plant growth, development, and physiological processes [19,20]. Consequently, the alleles assembled to modify flux in the biosynthetic pathway during the development of the maize inbred line with high provitamin A content may affect the production of carotenoids with unknown physiological functions that connect growth phases of plants with other complex metabolic processes [17]. The resulting changes in carotenoid profiles may then compete with other processes, leading to trade-offs between increased provitamin A content and agronomic performance in hybrids [20]. Studies conducted in maize reported that improvements in provitamin A content could affect grain yield of hybrid maize in a positive [21], negative [22,23], or neutral manner [24,25]. Khamkoh et al. [26] found simultaneous increases in lutein, zeaxanthin, and β-carotene as well as grain yield and its components in an orange waxy maize population improved through two cycles of recurrent selection. In another study examining the effect of genetic modification of provitamin A carotenoids on agronomic performance [20], a high-carotenoid transgenic corn line was similar to its near isogenic white corn line in terms of biomass, grain yield, and other agronomic traits under both controlled and field growing conditions. In contrast, Dhiliwayoa et al. [14] found that improvements in provitamin A, β-cryptoxanthin, and zeaxanthin concentrations through S1 recurrent selection were associated with grain yield reductions in two of the three target populations. These conflicting reports highlight the importance of testing provitamin A biofortified hybrids with diverse genetic backgrounds across a broad range of field environments to determine whether the assembly of favorable alleles to boost β-carotene content has any deleterious effect on agronomic performance of maize hybrids. This is an important step not only in identifying broadly adapted and commercially viable provitamin A biofortified hybrids, but also in adopting a breeding strategy that allows for more precise selection of new hybrids with high yield potential and enhanced concentrations of beneficial carotenoids for sustainable food and nutritional security. 

The biosynthesis and accumulation of carotenoids in crops are modulated both by the crop genotype and environmental factors [27,28]. Studies have demonstrated that many factors, including soil type, temperature, moisture, light intensity, occurrence of biotic and abiotic stresses, altitude, maturity cycle, and cropping practices significantly affect the synthesis and accumulation of individual carotenoids in maize and other crops [23,27,29,30,31,32,33]. Consequently, evaluating provitamin A biofortified maize hybrids involving diverse parents with medium to high provitamin A content, across variable climatic and other field growing conditions, is critical to elucidate the nature of the relationship between provitamin A accumulation and agronomic performance. Rigorous field testing of such biofortified hybrids, encapsulating the impacts of the current breeding scheme, is rarely done, but is necessary to assure the long-term success of delivering provitamin A biofortified maize hybrids for registration, release, and cultivation [13,20]. This is particularly important for smallholder farmers who frequently encounter food and nutritional insecurity and grow maize in the tropics under diverse rainfed conditions that can give rise to varying hybrid responses. The present study was therefore conducted across diverse growing environments to: (i) examine the relationship between the accumulation of provitamin A carotenoids and agronomic performance in hybrids, (ii) understand the changes in non-provitamin A carotenoids that occurred in biofortified hybrids, and (iii) determine the stability of the elevated levels of provitamin A and high grain yields in individual hybrids.

## 2. Results

### 2.1. Carotenoid Accumulation in Hybrids

In the covariance analyses, environment, hybrids, and hybrid by environment interactions had significant effects on all measured carotenoids, including provitamin A (Table 1). The heritability estimates for individual carotenoids varied from 0.88 to 0.95, indicating that the hybrid effect was stronger than the environmental and hybrid/environment interaction effects. On average, the proportion of each carotenoid found in hybrid grains was 24% for lutein, 41% for zeaxanthin, 14% for β-cryptoxanthin, 4% for α-carotene, and 17% for β-carotene. The hybrids exhibited a broad range of variation in concentrations of both provitamin A and non-provitamin A carotenoids (Appendix A). On average, the provitamin A biofortified elite (PVA) and commercial (COM-PVA) hybrids displayed increases of 20% in lutein, 1% in zeaxanthin, 22% in β-cryptoxanthin, 24% in α-carotene, and 76% in β-carotene in comparison with the orange commercial benchmark hybrid (OR-COM). Amongst all hybrids, 52 PVA and two COM-PVA hybrids had provitamin A concentrations varying from 10.0 to 14.0 µg/g (Appendix A), whereas the OR-COM hybrid had provitamin A content of 8.8 µg/g. The remaining hybrids exhibited provitamin A concentrations ranging from 7.6 to 9.9 µg/g. Many PVA hybrids with provitamin A content exceeding 10 µg/g were found to be comparable to the OR-COM in accumulating lutein and zeaxanthin (Appendix A).

All pair-wise genetic correlations among β-cryptoxanthin, α-carotene, and β-carotene were positive and strong (r_g_ = 0.98 to 0.99), whereas the correlations of lutein with zeaxanthin were weak (r_g_ = −0.07). Zeaxanthin had a strong and positive genetic correlation with β-cryptoxanthin (r_g_ =0.77) and α-carotene (r_g_ = 0.80) but had a strong negative correlation with β-carotene (r_g_ = −0.99). In contrast, lutein had a negative genetic correlation with β-cryptoxanthin (r_g_ = −0.33) and α-carotene (r_g_ = −0.23) but had a strong positive genetic correlation with β-carotene (r_g_ = 0.99). As the correlations among carotenoids in the present study did not always follow the expected relationships between carotenoids synthesized in the α- or β-branch of the biosynthetic pathway, we used canonical discriminant analysis to explore the formation of hybrid groups with similar carotenoid composition and content. The resulting first (CAN1) and second (CAN2) discriminant functions explained 73% and 27% of the total variations among hybrids, respectively (Table 2). CAN1 represented genetic changes favoring significant increases in accumulating zeaxanthin, β-cryptoxanthin, α-carotene, and β-carotene. In contrast, CAN2 was associated with genetic changes that encouraged a significant increase in accumulating lutein and zeaxanthin, but with significant decreases in accumulating β-cryptoxanthin and β-carotene. The dissimilarity in the composition of carotenoids that contributed the most to the two discriminant functions resulted in a significant (*p* < 0.0001) positive correlation of CAN1 with provitamin A (r = 0.86), but a negative correlation of CAN2 with provitamin A (r = −0.31). On the other hand, the correlations of total carotenoids with CAN1 (r = 0.86) and CAN2 (r = 0.40) were significant (*p* < 0.01) and positive. A scatter plot of CAN1 and CAN2 scores showed a clear separation of the 64 hybrids into three major groups with minimal overlaps (Figure 1). The first group (G-I) consisted of eight hybrids, all having negative CAN1 scores and most having positive CAN2 scores, which accumulated the lowest levels of zeaxanthin, provitamin A carotenoids, and total carotenoids, but moderate levels of lutein (Table 3). The second group (G-II) comprised of 36 hybrids, having a mixture of positive and negative CAN1 scores and mostly negative CAN2 scores (Figure 1), showing intermediate levels of zeaxanthin, provitamin A carotenoids, total carotenoids, and the lowest level of lutein relative to those in G-I. The third group (G-III) included 18 hybrids, having mainly positive CAN1 and CAN2 scores (Figure 1), and was characterized by the highest concentrations of both provitamin A and non-provitamin A carotenoids, as well as total carotenoids, in comparison to hybrids included in G-I and G-II (Table 3). 

### 2.2. Stability of Hybrids in Accumulating Provitamin A

A factor analytic (FA) model of order two (FA(2)), fitted for all carotenoids, and an order three (FA(3)), fitted for provitamin A, explained 87% to 95% of the hybrid by environment variation, showing an adequate fit of the model to our data sets. The FA model generated the latent regression line slopes for assessing the stability of the biofortified hybrids in accumulating provitamin A across varying growing conditions [34]. These slopes represented the responses of hybrids to varying field environments for each factor loading in the FA model, and the hybrids with larger slopes were considered stable. In the FA(3) model fitted for provitamin A, the estimated environment loadings for the first factor (FA1) were all positive, whereas those for the second (FA2) and third (FA3) factors had a mixture of positive and negative loadings. The latent regression line slopes for the first factor (FA1) was negatively correlated (r = −0.86, *p* < 0.0001) with those for the second factor (FA2), but was positively correlated (r = 0.61, *p* < 0.0001) with those for the third factor (FA3). Moreover, provitamin A content was positively correlated (*p* < 0.0001) with regression line slopes for FA1 (r = 0.67) and FA3 (r = 0.48), but was negatively correlated (*p* < 0.0001) with slopes for FA2 (r = −0.33). Since FA1 and FA2 represented 92% of the hybrid by environment variation, the latent regression line slopes for these factors provided a visual representation of the response patterns of the hybrids across environments (Figure 2). The slopes of these factors displayed a negative linear relationship accounting for 74% of the total variation in the response of hybrids to diverse growing environments. Amongst the biofortified hybrids, 15 in G-II and 11 in G-III combined positive slopes for FA1 with negative slopes for FA2, indicating that they accumulated increased provitamin A in environments characterized by both positive and negative estimated loadings (Appendix A). Sixteen hybrids among these also had positive slopes for FA3 varying from 0.3 to 2.5. As a result, these hybrids responded positively to all environments and accumulated consistently high levels of provitamin A. In contrast, seven of the eight hybrids in G-I combined negative slopes for FA1 and FA3 with positive slopes for FA2 and were thus accumulating less provitamin A in this set of environments. Several hybrids in G-II and G-III had slopes close to zero for FA1 and FA2, showing minimal or no responses to the test environments (Appendix A).

### 2.3. Agronomic Performance of Provitamin A Biofortified Hybrids in Diverse Field Environments

The agronomic traits used for combined analyses were recorded in 84 environments for ear height and 99 environments for grain yield. As shown in Table 4, environment, hybrid, and hybrid by environment interaction had significant effects on grain yield and other traits. The heritability estimates were strikingly high, ranging from 0.87 for ear height to 0.96 for anthesis and silking days. The best linear unbiased estimates (BLUPs) for yield varied from 2000 to 5111 kg/ha for PVA hybrids, from 3610 to 4567 for COM-PVA (H61 and H62) hybrids, 3051 kg/ha for OR-COM (H63) hybrids, and 3791 kg/ha for the farmer-preferred (H64) variety (Appendix A). Forty-two PVA hybrids, with provitamin A content exceeding 10 µg/g, produced 18 to 68% more grain yields than the OR-COM (H63) benchmark hybrid (Appendix A). Moreover, nearly 80% of these hybrids were found to be competitive with or better than the two COM-PVA (H61 and H62) hybrids in their yield potential. The genetic correlations between anthesis and silking days (r_g_ = 0.99), and between ear height and plant height (r_g_ = 0.98), were strong. On the other hand, the genetic correlation of anthesis and silking days with ear height and plant height were negative and weak (r_g_ < −0.25). Grain yield had negative genetic correlations with anthesis (r_g_ = −0.44) and silking days (r_g_ = −0.464), but had positive genetic correlations with ear height (r_g_ = 0.60) and plant height (r_g_ = 0.64). These results indicate that high yielding hybrids flowered and produced silks earlier, but tended to grow taller. The hybrid groups defined based on carotenoid profiles exhibited considerable differences in ear height, plant height, and grain yield (Table 5). On average, anthesis and silking days, as well as ear placement of hybrids, in G-I were comparable to those in G-II and G-III. On the other hand, hybrids in G-I were shorter and had lower average grain yield than those in G-II and G-III (Table 5). 

### 2.4. Yield Stability of Biofortified Maize Hybrids across Diverse Field Environments

A factor analytic model of order two (FA(2)) was fitted for anthesis and silking days, whereas that of order three (FA(3)) was fitted for grain yield, ear height, and plant height. These models accounted for 86% to 94% of the hybrid by environment variation. In the FA(3) model for grain yield, the first and second (FA1 and FA2) factor loadings jointly contributed to 84% of the genetic variation among hybrids, with the third factor (FA3) contributing an additional 10% of the variance. The latent regression line slopes for FA1 were positively correlated (*p* < 0.0001) with slopes for FA2 (r = 0.83), but negatively correlated (*p* < 0.0001) with slopes for FA3 (r = −0.52). Yield was weakly correlated (*p* = 0.0116) only with the latent regression lines slopes for FA3 (r = 0.31). As shown in Figure 3, the slopes for FA1 and FA2 exhibited a positive linear relationship, explaining 69% of the observed variation in the responses of hybrids to variable field environments. Four hybrids in G-I, 18 in G-II, and 9 in G-III had positive slopes for both FA1 and FA2 (Appendix A), indicating that they were more responsive to high yielding environments. Fourteen hybrids amongst these also had negative slopes for FA3, demonstrating that they responded positively to all favorable growing environments and produced consistently high grain yields. The remaining hybrids in the three groups had mainly negative regression line slopes for both FA1 and FA2 and were thus poorly adapted to the high yielding test environments (Figure 3). Very few hybrids had near zero regression slopes to be considered as nonresponsive hybrids to high yielding environments.

### 2.5. Trait Correlations for Carotenoids and Agronomic Traits 

Simple correlation analysis did not find significant relationships of individual provitamin A or non-provitamin A carotenoids with grain yield, ear height, and plant height (r = −0.14 to 0.20). Zeaxanthin was positively correlated (*p* < 0.0001) only with anthesis (r = 0.54) and silking days (r = 0.58). β-cryptoxanthin was weakly correlated (*p* < 0.05) only with silking days (r = 0.27), whereas α-carotene, β -carotene, and provitamin A were not significantly correlated with any of the agronomic traits. The latent regression line slopes for the first two factors of provitamin A were not significantly correlated with those for grain yield (r = −0.06 to 0.10), indicating that the response patterns of hybrids in accumulating provitamin A were not associated with their responses in producing grain yield. In fact, 14 biofortified hybrids had FA1 and FA2 latent regression line slopes showing favorable responses to environments that promoted accumulation of more than 10 µg/g of provitamin A and production of more than 3600 kg/ha grain yields (Appendix A). Seven hybrids amongst these (H08, H17, H23, H24, H41, H53, and H55) had latent regression line slopes for all three factor loadings in the FA(3) model showing favorable responses to all environments that encouraged accumulation of high provitamin A levels and production of high grain yields. Further comparisons of the best five high yielding PVA (TOP5PVA) hybrids with the orange endosperm commercial (OR-COM) benchmark hybrid showed that the former accumulated 47% more provitamin A and produced 35% more grain yield than the OR-COM hybrid (Figure 4). Moreover, the TOP5PVA hybrids were superior to the remaining PVA (OTHERPVA) and commercial PVA (COMPVA) hybrids in their yield potential as well as provitamin A content.

## 3. Discussion

### 3.1. Environmental Effects on Carotenoid Composition and Content

Enriching maize with provitamin A carotenoids through conventional breeding was advocated as a viable approach for adding vitamin A directly to the diets of consumers who rely on predominantly starchy foods with limited access to fruits and vegetables [10]. The biosynthesis and accumulation of provitamin A carotenoids in plants, including maize, is controlled by the genetic makeup of the variety and its response to environmental stimuli to meet crop development requirements [30,35]. Consequently, changes in the composition and content of provitamin A carotenoids under the influence of varying environmental factors may lead to a trade-off with grain yield and other traits due to the potential competition for precursors and energy [20]. We therefore conducted the present study to determine the relationship between accumulating varying levels of provitamin A and agronomic performance of biofortified maize hybrids across variable environmental conditions in West Africa. Environment, hybrid, and hybrid by environment interaction had significant effects on individual carotenoid accumulation in our study, possibly due to changes in enzymatic activity in the carotenoid biosynthetic pathway triggered by the variable temperature, light intensity, humidity, precipitation, physical and chemical soil properties, and the occurrence of biotic and abiotic stresses encountered in the field during the testing of the hybrids over a period of four years [27,36,37,38,39]. Nonetheless, the high heritability estimates found for individual carotenoids suggested that the variations in carotenoid synthesis and accumulation were largely regulated by the genetic component rather than by environmental effects [30]. Similar results were also reported in studies that explored the extent of variation in carotenoid composition and content in diverse maize germplasm [22,32,40,41,42,43,44,45,46]. Our study demonstrated that biofortified maize hybrids accumulating high levels of individual carotenoids could be developed for cultivation under a broad range of crop management practices [47] and growing conditions, notwithstanding the importance of environmental and interaction effects on carotenoid biosynthesis.

### 3.2. Changes That Occurred in Pro-Vitamin A and Non-Provitamin A Carotenoids

The biofortified maize hybrids included in our study displayed considerable variation in accumulating different amounts and types of carotenoids in their grains, likely due to the diversity of alleles, derived from their parents, which regulate the differential expression of structural genes in the carotenoid biosynthetic pathway [12]. Nearly 80% of these hybrids accumulated 20 to 66% more provitamin A in their grains in comparison to the orange endosperm commercial hybrid. Lutein and zeaxanthin remained the major carotenoid fractions in the kernels of the biofortified hybrids despite increases in β-cryptoxanthin, α-carotene, and β-carotene content. Similar results of the predominance of lutein and zeaxanthin associated with increases in concentrations of β-cryptoxanthin, α-carotene, and β-carotene were reported earlier in biofortified maize inbred lines [13,48] and hybrids [43,49]. Many studies involving diverse maize inbred lines and hybrids [22,42,43,44,46,50,51] also documented lutein and zeaxanthin as constituting more than 75% of the total carotenoids in their grains. In contrast, inbred lines and hybrids containing the favorable *crtRB1* allele had provitamin A carotenoids constituting 49 to 86% of the total carotenoids, mainly due to the associated increase in the flux of precursors to β-carotene [33,52]. The current study highlighted the significant progress made in provitamin A enrichment of hybrids without compromising the levels of other carotenoids with beneficial properties. Consumption of diets rich in these carotenoids can provide multiple health benefits, including improved vision, boosted immune responses, reduced risks for the onset of age-related macular eye disease and cataracts, cardiovascular diseases, and cancer, particularly in populations that depend on maize as a major food source [53,54,55,56,57,58].

The variation in the carotenoid profiles in maize genotypes grown in diverse locations and seasons is regulated by many loci distributed across three metabolic pathways that display additive as well as pleiotropic effects [59]. In the present study, the provitamin A biofortified maize hybrids were separated into three major groups with distinct carotenoid composition and content. The hybrids in the second and third groups accumulated elevated levels of provitamin A carotenoids as well as zeaxanthin, as opposed to those in the first group that displayed the lowest levels of provitamin A carotenoids and zeaxanthin. Studies in *Arabidopsis* and tobacco found that over-expression of the β-carotene hybroxylase gene significantly increased zeaxanthin, which markedly increased the flux of the xanthophyll cycle [60,61]. Although accumulation of lutein did not follow any specific trend in the three hybrid groups, its content did not differ markedly among the hybrid groups. In addition, increases in provitamin A content was significantly correlated with increases in the total carotenoids in hybrids. It is thus likely that parental lines contributed allelic variants at the *PSY* and *ZDS* loci, inducing increases in the synthesis of metabolites upstream in the biosynthetic pathway that enhanced substrate flux to both the α- and ß-branches downstream in the pathway, leading to higher concentrations of carotenoids with provitamin A activity, while at the same time maintaining appreciable levels of other carotenoids with health benefits. These results suggest that simultaneous increases in concentrations of both provitamin A and non-provitamin A carotenoids could be attained in maize grains by maximizing the total carotenoid synthesis and accumulation, consistent with the results reported in sweet and waxy corn [26,32]. This may be achieved through visual selection for darker orange kernel color, which is associated with higher total carotenoids [45,62], followed by the selection for favorable alleles of crtRB1-3T’ and crtRB1-5′TE to enhance substrate flux for more synthesis of carotenes [48,63,64].

### 3.3. Adaptability of PVA Hybrids across Diverse Growing Environments

The stability of provitamin A levels in hybrids across locations and seasons is critical for the commercial release of biofortified hybrids in areas where smallholder farmers cultivate maize under varying soil and climatic conditions and employ different crop management practices. In the present study, the factor analytics model was effective in capturing the stability of accumulating provitamin A across variable growing environments. We found some hybrids displaying elevated accumulation of provitamin A in response to both the first factor and second factor environment loadings. It thus appears that these hybrids had minimal cross-over type of interaction with diverse environments that allowed consistently high levels of expression of provitamin A in their grains [65]. In contrast, other hybrids exhibited either negative or neutral responses to the two environment factor loadings. Although Cullis et al. [66] considered hybrids with close to zero latent regression line slopes as stable products because of their insensitivity to changes in environment loadings, some hybrids with favorable responses to both factor loadings in our study can be considered stable for high provitamin A content across all environments, as recommended by Zhang et al. [67]. Such stable and broadly adapted biofortified maize hybrids can be successfully commercialized to farmers to provide guaranteed nutrient content in areas where vitamin A deficiency is severe.

### 3.4. Grain Yield of PVA Hybrids across Diverse Growing Environments

The acceptability of biofortified hybrids by farmers depends on combining high levels of provitamin A with high yield potential and other desirable agronomic features, which are complex traits with polygenic inheritance that are significantly affected by environmental factors [8,27,28]. In the current study, many elite provitamin A biofortified hybrids produced grain yields exceeding the yield of the orange endosperm commercial benchmark hybrid by as high as 68%. Some of these hybrids were also found to be as high yielding as or higher yielding than the two commercial provitamin A biofortified hybrids, demonstrating the potential to achieve concurrent improvements in provitamin A content and grain yield in maize hybrids. The observed weak correlations of both provitamin A and non-provitamin A carotenoids with grain yield and other agronomic traits further support the feasibility of selecting elite parental lines with high provitamin A content for developing productive biofortified hybrids that farmers require to profitably produce maize.

Apart from high yield potential, farmers are also interested in hybrid yield stability across seasons to minimize the risks of crop failure, when the growing seasons are unfavorable, and benefit from harvesting more grain, when the growing seasons are favorable. In the factor analytic model (FA(3)), the hybrids displayed positive, neutral, and negative latent regression line slopes for grain yield, indicating the diversity of responses of the PVA hybrids to prevalent changes in field growing conditions. Some PVA hybrids, showing positive slopes for the first two factors, responded favorably to the diversity of growing environments and produced consistently high grain yields. These hybrids may take full advantage of suitable growing seasons with adequate rainfall and sunlight as well as favorable temperature and humidity that enhance photosynthesis [68]. Such adapted hybrids may have more plastids, including chloroplasts and chromoplasts, when they are grown under favorable growing seasons for carotenoid biosynthesis [69]. Dannehl et al. [70] demonstrated that tomatoes grown under optimum climatic conditions had increased photosynthesis and ß-carotene concentration in their fruits, suggesting that enhanced photosynthesis provided the precursors for biosynthesis of ß-carotene. It is therefore reasonable to assume that conventional breeding can successfully combine the benefits of high yield potential with stable expressions of provitamin A levels in maize hybrids. 

### 3.5. Effect of Accumulating PVA Carotenoids on Agronomic Performance of Hybrids

Considerable heterogeneity in genetic variances were found for provitamin A content and grain yield, indicating the diversity of the environments encountered during the evaluation of the hybrids in the present study. Despite these growing conditions, the correlations of provitamin A and other carotenoids with grain yield were desirable with no apparent deleterious effects on the agronomic performance of the tested PVA hybrids. We found superior hybrids with stable expression of high provitamin A and high grain yields across diverse growing conditions, consistent with the results of another study that reported no relationship between carotenoid components and grain yields [52]. Muthusamy et al. [71] developed many hybrids, combining high provitamin A content with high grain yields that were competitive to their original versions as well as normal hybrid checks. Our study demonstrated the effectiveness of rigorous selection of hybrids for agronomic performance and adaptive traits through successive testing stages in developing provitamin A biofortified maize hybrids with high yield potential and broad adaptation across diverse growing environments. In this regard, a breeding strategy, involving vigorous selection of maize inbred lines for desirable agronomic and adaptive traits, followed by selection for elevated levels of beneficial carotenoids, can facilitate the development of maize hybrids with superior agronomic performance and much higher levels of provitamin A and other beneficial carotenoids that are broadly adapted to the diverse tropical production environments in Africa. 

## 4. Materials and Methods

### 4.1. Genetic Materials

Maize inbred lines of diverse origin with intermediate to high levels of provitamin A were evaluated, in hybrid combinations in multiple locations through successive stages, until provitamin A biofortified hybrids with desirable agronomic performance were identified for dissemination to partners for regional testing. A set of regional trials, consisting of 30, 30, 36, and 36 hybrids were evaluated in 2015, 2016, 2017, and 2018, respectively, under rainfed conditions across many locations in five countries (Appendix A). New hybrids were added every year, whereas inferior ones were removed from these trials, leading to year-to-year variation in the number of tested hybrids. In total, 60 provitamin A biofortified (PVA) single-cross and three-way cross hybrids (H01-H60), two commercial provitamin A biofortified (COM-PVA) single cross hybrids (H61-H62) in Nigeria, and an orange endosperm commercial (OR-COM) single-cross hybrid (H63) marketed extensively in Nigeria for many years, as well as a local maize (LV) variety (H64) were included in the regional trials (Appendix A). The 60 PVA hybrids were formed from maize inbred lines derived from backcrosses containing temperate germplasm as donors of high β-carotene, as well as bi-parental crosses of elite high provitamin A lines. The genetic backgrounds of the temperate donor lines and the development of lines with intermediate to high levels of provitamin A from backcrosses were extensively described by Menkir et al. [13]. The two COM-PVA hybrids in Nigeria and the OR-COM hybrid that was not bred specifically for high provitamin A content, obtained from Premier Seeds Nigeria Ltd., were included as benchmarks. Farmer-preferred recycled hybrids or improved open-pollinated maize varieties commonly grown around the testing sites where the regional trials were conducted were added to the regional trials by partners as a local maize variety check (LV). In the IITA testing sites where the regional trials were conducted in Nigeria, the OR-COM hybrid was once again used as the farmers-preferred hybrid to obtain a more reliable measurements of carotenoids in this benchmark hybrid for subsequent comparisons with the PVA hybrids. Amongst the 64 hybrids included in the regional trials, 10 were tested for four years, 9 were tested for three years, 12 were tested for two years, and the remaining 33 were tested for one year only.

### 4.2. Performance Evaluation in Multi-Environment Trials (MET) 

The 30, 30, 36, and 36 hybrids included in the regional trials were arranged in 5 × 6, 5 × 6, 6 × 6, and 9 × 4 alpha lattice designs, respectively, and were evaluated with three replications during the main rainy seasons in collaboration with partners in the national agricultural research systems (NARS) and private seed companies in 23 locations in 2015, 28 locations in 2016, 32 locations in 2017, and 20 locations in 2018 in Benin Republic, Cameroon, Ghana, Mali, and Nigeria. These test locations represent the diverse maize growing environments and agro-ecological zones stretching throughout West and Central Africa (Appendix A). Each hybrid was planted in a single 5 m long row with spacing of 0.75 m between rows and 0.5 m between plants within a row. At the IITA experiment stations in Nigeria, two seeds were planted in each hill and were later thinned to one plant after emergence to attain a population density of 53,000 plants per ha. At the time of sowing, we applied 60 kg N, 60 kg P, and 60 kg K ha^−1^ fertilizer with an additional 60 kg N ha^−1^ fertilizer applied four weeks later. The trial fields were sprayed with gramazone and atrazine as pre-emergence herbicides at the rate of 5 L ha^−1^ and were followed by manual weeding to keep the trials weed-free. The collaborators in the national agricultural research systems (NARS) and private seed companies used crop management practices, rates of fertilizer application, and weed control methods recommended for each of their testing location when they conducted these trials.

### 4.3. Agronomic Trait Measurements

Days to anthesis and silking were recorded in each plot as the number of days from planting to when 50% of the plants were shedding pollen and showed emerged silks, respectively. Plant and ear heights were measured in cm as the distance from the base of the plant to the height of the first tassel branch and the node bearing upper ear, respectively. All ears harvested from each plot were shelled to determine percent moisture, which was used to determine grain yield, adjusted to 15% moisture. Grain yield was calculated from ear weight and grain moisture, assuming a shelling percentage of 80% and final adjusted moisture content of 15% in each testing site.

### 4.4. Analysis of Carotenoids

To avoid contamination from pollen originating from other maize hybrids, four representative plants were self-pollinated from each hybrid and cobs were harvested with the husk in the first two replications at Ibadan, Ikenne, Kadawa, Mokwa, Saminaka, and Zaris in Nigeria from 2015 to 2018. These locations represent the humid forest, moist savanna, and dry savanna agro-ecological zones across West and Central Africa [72]. Cobs were not harvested at Mokwa in 2016 due to severe drought that adversely affected seed setting in hybrids. The cobs were carefully threshed to form composite samples for carotenoid analysis at the Crop Utilization Laboratory of the International Institute of Tropical Agriculture using HPLC. The extraction protocol from grain samples and subsequent carotenoid analyses were carried out using the procedure described by Howe and Tanumihardjo [73]. Provitamin A was calculated for each sample as the sum of β-carotene (all-trans plus 13-cis and 9-cis isomers) plus 50% each of α-carotene and β-cryptoxanthin. 

### 4.5. Statistical Analysis

The yearly addition of new hybrids and the removal of inferior ones in the four regional trials created unbalanced data sets for carotenoids and agronomic traits recorded in these trials. Year-location combinations are hereafter referred to as environments. The combined data across location x year (environment) for carotenoids and agronomic traits were analyzed using the mixed model approach [74,75], with a factor analytic (FA) structure, to model the effects of hybrids and hybrid by environment interactions effectively and increase the precision of predicting breeding values [34]. The first step in the analysis involved separate estimation of genetic variances from the data recorded in each environment. All environments with zero genetic variance estimates were excluded from the analysis. A combined analysis across all environments was then computed for each carotenoid or agronomic trait using a liner mixed model following this formula:$$y=Χ\tau +{Ζ}_{g}u_{g}+{Ζ}_{\eta }u_{\eta }+e$$
where τ is a p-vector of fixed effects and of environment means, with the associated design matrix Χ. The vector $u_{g}$ is the (m×p)-vector of random genetic effects, ordered as genotypes within environments, with the associated design matrix ${Ζ}_{g}$. The vector $u_{\eta }$ is of the random non-genetic effects (related to the experimental designs of individuals environments) with the associated design matrix ${Ζ}_{\eta }$. The vector of residuals is given by e. We assume that $u_{g}$, $u_{\eta },$ and e are mutually independent and distributed as Gaussian, with zero means, such that: $u_{g}∽Ν\left(0,G_{g}\right)$, $u_{\eta }∽Ν\left(0,G_{\eta }\right)$, and e∽Ν(0,R).

In the combined analysis, all effects (replicate nested within environments, hybrids, environment and the interaction between genotype and environment) were considered random and the residual variances were assumed to be heterogeneous. The hybrids were considered as random because they represent the diversity of elite provitamin A biofortified hybrids developed in our breeding program and disseminated to the partners for extensive testing. The variance components estimated from the data, combined across environments, were computed using the restricted maximum likelihood (REML) procedure, and the significance of the variances was determined with a residual maximum likelihood ratio test. We ran a full model with all random effects and a reduced model without that specific random effect. The difference between the log-likelihoods of the full and reduced models followed a chi-square distribution with one degree of freedom. Heritability estimates were calculated for each trait using the method of Cullis et al. [65] that accommodate unbalanced datasets. 

Factor analysis, which has become a standard model for analyzing data recorded in multi-environment trials, was employed to analyze carotenoid content and agronomic traits using the genetic variance-covariance matrix so that:$$u_{g}=\left(L_{g}⊗I_{m}\right)f_{g}+\delta_{g}$$
where $L_{g}$ is the p×2 matrix of loadings (environment effects); $f_{g}$ is the (m×2)-vector of scores (genotype effects), and $\delta_{g}$ is the (m×p)-vector of lack of fit effects. It is assumed that: $f_{g}∽Ν\left(0,\;1\right)$ and $\delta_{g}∽Ν\left(0,\psi_{g}\right)$.

An FA model was then a (multiple) regression of genotype by environment effects on environment covariates (loadings) with separate slopes (scores) for each genotype. The REML estimates of the factor loadings were rotated through a principal component solution to determine the percent variance accounted for by the factors and estimate the hybrid scores following the procedures of Smith and Cullis [76]. The algorithm used in ASReml-R set the elements of the upper triangle of $L_{g}$ to zero [77]. More details on the model fitting, estimation, and prediction can be found in Gogel et al. [75] and Smith and Cullis [76]. 

As provitamin A and grain yield were targeted as primary traits in the present study, we fitted the factor analytic model of order three to determine hybrid stability across varying growing conditions, through the latent regression analysis of predicted breeding values on rotated environmental loadings of each of the three factors [34]. This provided graphical representation of the hybrid by environment effects from an FA model. All analyses were carried out in R [77] using ASReml-R (v4.1) for fitting the mixed models [78]. Simple correlation analysis was then computed between the latent regression line slopes for each of the three factors for provitamin A and those of grain yield to assess the relationship of the two primary traits measured across diverse environments.

As the variances were heterogeneous for these traits in each environment, multivariate analysis was conducted using ASReml-R to estimate the genotypic correlations among carotenoids or agronomic traits. Moreover, simple correlation analysis of provitamin A content with grain yield and other agronomic traits of hybrids was conducted using PROC CORR in SAS [79]. To determine the carotenoid profiles of the hybrids, the best linear unbiased estimates (BLUPs) of the five carotenoids, excluding that of provitamin A, were subjected to principal component analysis using the correlation matrix. The resulting five principal component axes (PC1-PC5) scores were then used to stratify the hybrids into groups using Ward’s [80] clustering method. The hybrid groups, plus the PC1 to PC5, axes scores were then used to run canonical discriminant analyses, following the CANDISC procedure in SAS [79]. This analysis was run because it was effective in providing a clear separation of hybrids into groups based on the first two factors. Simple correlation analysis between hybrid BLUPs for carotenoids, and the corresponding hybrid scores for the two canonical discriminant functions (CAN1 and CAN2), were conducted to identify carotenoids significantly contributing to each function. Descriptive statistics were computed for individual carotenoids and agronomic traits of hybrid groups using the univariate procedure in SAS [79].

## 5. Conclusions

Using many biofortified maize hybrids with diverse genetic backgrounds, we demonstrated that the variation in accumulating provitamin A, as well as other carotenoids, with additional health benefits was primarily regulated by the genetic makeup of the hybrids when measured across diverse field growing environments. Many hybrids accumulating more than 10 µg/g of provitamin A and producing more than 3600 kg/ha grain yields, relative to the orange commercial maize hybrid (3051 kg/ha), were found to be as high yielding as, or higher yielding, than the commercial biofortified hybrids. Most of these hybrids were competitive to the orange commercial maize hybrid in accumulating lutein and zeaxanthins with additional health benefits. Our study demonstrated that breeding for enhanced provitamin A content had no adverse effect on production potential and agronomic performance of hybrids evaluated in the regional trials. Considering the importance of carotenoids to human nutrition and health, and the increasing demand from consumers for healthy foods, the development and delivery of maize hybrids with much higher levels of provitamin A and other important carotenoids can maximize their health benefits [12,75,81,82,83] by offsetting the potential nutrient losses resulting from the diversity of storage practices, methods of milling grains, and preparations of traditional foods. Some outstanding biofortified hybrids identified in the present study (H08, H17, H23, H24, H41, H53, and H55) with stable expressions of high provitamin A and high yield potential are suitable candidates for commercialization in tropical lowland agroecological zones to enhance productivity and yield stability for smallholder farmers. Furthermore, our study highlighted the possibility of developing broadly adapted hybrids that are attractive to private seed companies for commercialization across a wide range of maize production conditions. 

## Figures and Tables

**Figure 1 plants-10-01580-f001:**
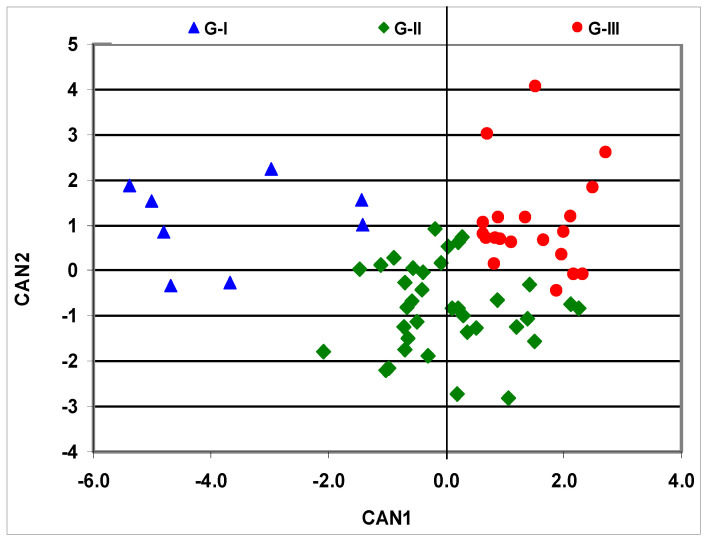
Scatter plot of the two canonical discriminant functions (CAN1 and CAN2) for carotenoids of maize hybrids, with those marked blue belonging to the first group (G-I), hybrids marked green included in the second group (G-II), and hybrids marked red representing the third group (G-III), that were measured in regional trials evaluated across 23 test environments from 2015 to 2018.

**Figure 2 plants-10-01580-f002:**
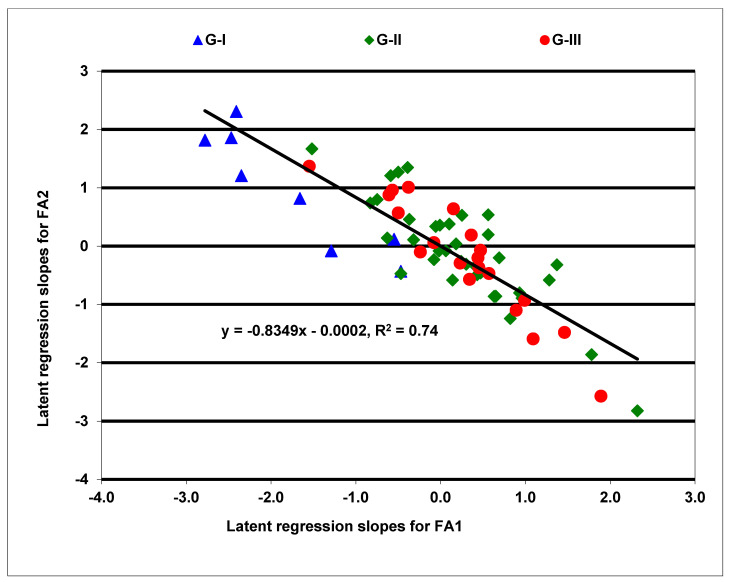
Latent regression line slopes of the first two factors for provitamin A content of maize hybrids were obtained from a factor analytic model. Those marked blue belong to the first group (G-I), those marked green are included in the second group (G-II), and those marked red represent the third group (G-III).

**Figure 3 plants-10-01580-f003:**
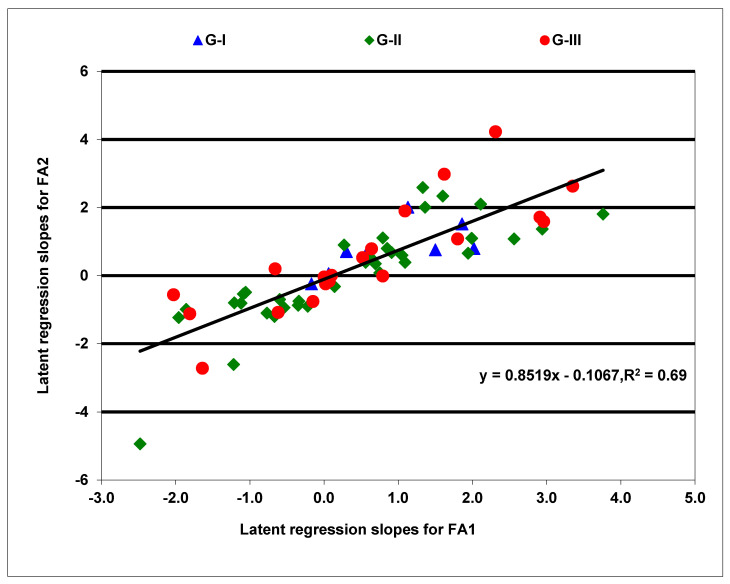
Latent regression line slopes of the first two factors for grain yield of maize hybrids were obtained from a factor analytic model. Hybrids marked blue belong to the first group (G-I), those marked green represent the second group (G-II), and those marked red were included in the third group (G-III).

**Figure 4 plants-10-01580-f004:**
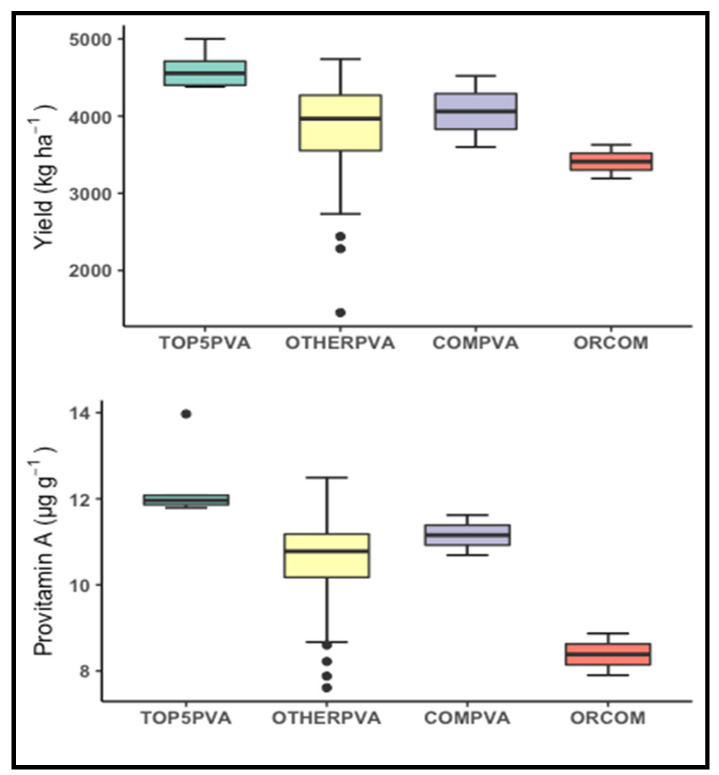
Box-plots of provitamin A content and grain yields of the five best provitamin A biofortified (TOP5PVA), the remaining provitamin A biofortified (OTHERPVA), commercial provitamin A biofortified (COMPVA), and an orange commercial (ORCOM) hybrids recorded across diverse environments.

**Table 1 plants-10-01580-t001:** Covariance estimates and their significant test for carotenoids, including provitamin A, through the residual maximum likelihood ratio test (REMLRT) based on the combined analyses of trials involving 64 hybrids evaluated across 23 environments.

Carotenoids	Covariance and REMLRT	Hybrid	Environment	Environment × Hybrid
Lutein	Covariance	2.8 ****	9.4 ****	0.8 ****
	REMLRT	217.2	55.6	35.3
Zeaxanthin	Covariance	13.7 ****	9.2 ****	0.5 ****
	REMLRT	331.9	15.4	39.0
β-cryptoxanthin	Covariance	1.7 ****	1.3 ****	0.1 ****
	REMLRT	415.0	27.8	45.8
α-carotene	Covariance	0.1 ****	0.1 ****	0.003 ****
	REMLRT	317.0	8.7	38.1
β- carotene	Covariance	1.7 ****	1.9 ****	0.1 ****
	REMLRT	383.4	15.3	52.2
Provitamin A	Covariance	3.7 ****	3.3 ****	0.2 ****
	REMLRT	465.4 ****	27.8 ****	47.8 ****

**** Significant *p* < 0.0001 levels using residual maximum likelihood ratio test (REMLRT).

**Table 2 plants-10-01580-t002:** Simple correlations between coefficients of each canonical discriminant function (CAN1 and CAN2) and the best linear unbiased predictors (BLUPs) of individual carotenoids, including provitamin A of 64 hybrids evaluated across 23 environments.

	Simple Correlation Coefficients with	
Carotenoids	CAN1	CAN2
Lutein	0.18	0.93 ****
Zeaxanthin	0.58 ****	0.27 *
β-cryptoxanthin	0.68 ****	−0.27 *
α-carotene	0.80 ****	−0.09
β-carotene	0.84 ****	−0.33 **
Provitamin-A	0.86 ****	−0.31 *
Variance (%)	0.73	0.27
Canonical correlations (CC)	0.84	0.69
Significant levels for CC	*p* < 0.0001	*p* < 0.0001

*, **, **** Significant at *p* < 0.05, *p* < 0.01, and *p* < 0.0001 levels, respectively.

**Table 3 plants-10-01580-t003:** Minimum, maximum, and mean carotenoid concentrations for three groups of hybrids (G-I, G-II, and G-III) defined based on canonical discriminant analyses.

Carotenoids	Minimum	Maximum	Mean ± SE
	G-I		
Lutein (µg/g)	8.3	11.8	10.5 ± 0.5
Zeaxanthin (µg/g)	13.9	17.3	15.5 ± 0.4
β-cryptoxanthin (µg/g)	4.0	5.4	4.7 ± 0.2
α-carotene (µg/g)	1.1	1.5	1.3 ± 0.1
β-carotene (µg/g)	4.1	6.0	5.0 ± 0.2
Total Carotenoids (µg/g)	34.7	41.9	37.0 ± 1.0
Provitamin-A (µg/g)	7.6	9.5	8.4 ± 0.2
	G-II		
Lutein (µg/g)	7.7	11.1	9.1 ± 0.1
Zeaxanthin (µg/g)	13.7	18.5	16.7 ± 0.2
β-cryptoxanthin (µg/g)	4.2	7.1	6.1 ± 0.1
α-carotene (µg/g)	1.2	1.9	1.6 ± 0.0
β-carotene (µg/g)	5.9	8.8	7.0 ± 0.1
Total Carotenoids (µg/g)	36.2	43.8	40.6 ± 0.3
Provitamin-A (µg/g)	9.8	12.5	10.9 ± 0.1
	G-III		
Lutein (µg/g)	9.3	15.0	11.3 ± 0.3
Zeaxanthin (µg/g)	15.2	20.5	17.9 ± 0.3
β-cryptoxanthin (µg/g)	5.1	8.4	6.2 ± 0.2
α-carotene (µg/g)	1.5	2.1	1.7 ± 0.0
β-carotene	6.5	8.0	7.3 ± 0.1
Total Carotenoids (µg/g)	40.8	48.9	44.4 ± 0.5
Provitamin-A (µg/g)	10.1	14.0	11.2 ± 0.2

**Table 4 plants-10-01580-t004:** Covariance estimates and their significant test for agronomic traits through the residual maximum likelihood ratio test (REMLRT) based on the combined analyses across environments.

Traits	Covariance and REMLRT	Hybrid	Environment	Environment × Hybrid
Anthesis days	Covariance	1.4 ****	24.8 ****	0.7 ****
	REMLRT	294	601	385
Silking days	Covariance	1.4 ****	23.3 ****	0.8 ****
	REMLRT	288	560	502
Ear height	Covariance	12.6 ****	452.4 ****	13.8 ****
	REMLRT	425	411	209
Plant height	Covariance	30.2 ****	873.6 ****	36.0 ****
	REMLRT	462	470	217
Grain yield	Covariance	496,206 ****	1,689,020 ****	406,147 ****
	REMLRT	567	337	677

**** Significant *p* < 0.0001 levels using residual maximum likelihood ratio test (REMLRT).

**Table 5 plants-10-01580-t005:** Minimum, maximum, and mean agronomic traits for the four groups of hybrids (G-I, G-II, and G-III) defined based on canonical discriminant analyses of carotenoids.

Traits	Minimum	Maximum	Mean
	G-I		
Anthesis days	58	62	59 ± 0.4
Silking days	60	64	62 ± 0.5
Ear height (cm)	85	101	90 ± 1.9
Plant height (cm)	159	183	176 ± 2.9
Grain yield (kg/ha)	3245	4559	3846 ± 168
	G-II		
Anthesis days	58	61	59 ± 0.1
Silking days	60	64	61 ± 0.2
Ear height (cm)	85	95	90 ± 0.4
Plant height (cm)	176	193	183 ± 7.0
Grain yield (kg/ha)	2254	5054	3977 ± 103
	G-III		
Anthesis days	58	62	60 ± 0.2
Silking days	60	65	62 ± 0.2
Ear height (cm)	82	95	89 ± 0.4
Plant height (cm)	171	192	181 ± 1.2
Grain yield (kg/ha)	1420	4812	3884 ± 176

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
