# Peer review of "Unravelling the Effect of Provitamin A Enrichment on Agronomic Performance of Tropical Maize Hybrids"

_plants, 2021, doi:10.3390/plants10081580_

Round 1
Reviewer 1 Report
Manuscript ID: plants-1268362
Manuscript title : Unravelling the Effect of provitamin A Enrichment on Agronomic Performance of Tropical Maize Hybrids
Authors: Abebe Menkir, Ibnou Dieng, Wende Mengesha, Silvestro Meseka, Bussie Maziya-Dixon, Oladeji Emmanuel Alamu, Bunmi Bossey, Oyekunle Muhyideen, Manfred Ewool and Mmadou Coulibaly
The present manuscript describes a medium-term study aimed at finding the best combinations among high contents of provitamin A and desirable agronomic traits in maize hybrids cultivated in different agroecological zones of the savanna type in West and Central Sub-Saharan Africa.
The rationale behind the study appears to be well aimed and clear, the results are expected to be useful not only for farmers but also for a variety of public and private stakeholders; the amount of work done is huge, the manuscript is mostly well written, even if a thorough revision of language and editing would be highly beneficial, in general, and in particular (just as examples taken from the first pages):
- at line 47, where “Countries” and not “countries” should be written
- at line 49, where [10-13] and not [10, 11, 12, 13] should appear
- at line 64, where “photo-oxidative” and not “phyto-oxidative” should be written
- at line 68, where the meaning of the word “continuing” is hard to understand, in that context
…..and so forth…
Last but not least, the Authors should refrain from the compulsive using, in the text, in the figures/tables, AND in the supplementary materials, of uncommon/specialistic acronyms without defining them in full upon first mention
The only potential, albeit critical, problem I see with the present manuscript is that, because of its content, goals, approach and methodology, it would seem not particularly suited for “Plants”. Instead, it would be ideally suited for a more agronomy/crop physiology-oriented journals…For example, “Agronomy”, another well-reputed journal of the MDPI Publisher.
In summary, I recommend minor revision, whereas I leave in the hands of the Editor the decision whether to suggest to the Authors another journal as the “natural” and more logical archive for the present manuscript. In any case, I would gladly evaluate a revised version of it, taking into account the points raised above.
Author Response
Reviewer 1
Manuscript ID: plants-1268362
Manuscript title : Unravelling the Effect of provitamin A Enrichment on Agronomic Performance of Tropical Maize Hybrids
Authors: Abebe Menkir, Ibnou Dieng, Wende Mengesha, Silvestro Meseka, Bussie Maziya-Dixon, Oladeji Emmanuel Alamu, Bunmi Bossey, Oyekunle Muhyideen, Manfred Ewool and Mmadou Coulibaly
The present manuscript describes a medium-term study aimed at finding the best combinations among high contents of provitamin A and desirable agronomic traits in maize hybrids cultivated in different agroecological zones of the savanna type in West and Central Sub-Saharan Africa.
The rationale behind the study appears to be well aimed and clear, the results are expected to be useful not only for farmers but also for a variety of public and private stakeholders; the amount of work done is huge, the manuscript is mostly well written, even if a thorough revision of language and editing would be highly beneficial, in general, and in particular (just as examples taken from the first pages):
- at line 47, where “Countries” and not “countries” should be written - Done
- at line 49, where [10-13] and not [10, 11, 12, 13] should appear - Done
- at line 64, where “photo-oxidative” and not “phyto-oxidative” should be written - Done
- at line 68, where the meaning of the word “continuing” is hard to understand, in that context
…..and so forth…– Replaced with the correct word as suggested.
Last but not least, the Authors should refrain from the compulsive using, in the text, in the figures/tables, AND in the supplementary materials, of uncommon/specialistic acronyms without defining them in full upon first mention - The acronyms were written in full where every they appeared first followed by subsequent use of the acronyms in brackets
The only potential, albeit critical, problem I see with the present manuscript is that, because of its content, goals, approach and methodology, it would seem not particularly suited for “Plants”. Instead, it would be ideally suited for a more agronomy/crop physiology-oriented journals…For example, “Agronomy”, another well-reputed journal of the MDPI Publisher. - The manuscript focuses on the effect of changes in carotenoids, which are essential metabolites for photosynthesis, growth, development and defense against biotic and abiotic stresses in plants that make it suitable for publication in the special issue of Biosynthesis and Function of Plant Specialized Metabolites.
In summary, I recommend minor revision, whereas I leave in the hands of the Editor the decision whether to suggest to the Authors another journal as the “natural” and more logical archive for the present manuscript. In any case, I would gladly evaluate a revised version of it, taking into account the points raised above.
Reviewer 2 Report
Abstract or elsewhere in the text:
What types of hybrids were the provitamin A biofortified hybrids? Single cross, three-way or four-way hybrids. Please enlist and distinguish these in the Supplementary Table 1.
What are the field environments? Are these field conditions of multiple sites or seasons?
Provide some values/figures related to higher grain yields for the new hybrids (e.g. > xx t/ha) that accumulated more than 10 µg/g of provitamin A compared with the existing orange commercial maize hybrid hybrids (e.g. > xx t/ha).
Overall, which of the hybrids were recommended for providing higher grain yields, higher provitamin A and for being stable performers?
The following conclusion is debatable:-
‘Our study showed that breeding for enhanced provitamin A content had no adverse effect on grain yield in biofortified hybrids.’ Grain yield is often negatively correlated with enhanced nutritional content. Hence, low yields in biofortified varieties are the main cause for low adoption rate. Perhaps authors should reposition their conclusion to refer to ‘the present hybrid populations and field environments’ rather than providing general conclusions.
Introduction
Authors have provided extensive review on carotenoids and their bio-synthesis and accumulation. This should be kept to a minimum since the focus is to elucidate agronomic performance of provitamin A enriched tropical maize hybrids under multiple field conditions.
Authors should provide sufficient background on the following aspects:
historical perspectives on breeding and genetic analyses for provitamin A enrichment; progress on variety development, registration and release of provitamin A enriched tropical or temperate maize hybrids; bottlenecks on adoption rate of biofortified maize; the influence of production environments on the expression of provitamin A, yield and yield related traits; correlations between yield and yield related quantitative traits and provitamin A and other nutrient contents. Any findings that reported breeding for enhanced provitamin A or nutrient content and its effect on grain yield in hybrids or OPVs.
Before the objective statement, authors should relate this work to their current breeding efforts and the need and the cases for assessing provitamin A enrichment and agronomic performance under multiple field environments in their hybrid breeding program for recommendation, hybrid breeding etc.
Results
It will be interesting if authors can provide the mean values and some statistical contrasts amongst the tested hybrids for grain yield and nutrient profiles. The mean values should be cascaded by the test countries or selected regions/agro-ecologies and years of evaluation. I think one can make several inferences from this data set. Please provide this in a Supplemental Table.
Table 1.
Please specify the number of test environments and hybrids involved in this analysis.
Provide variable names for the columns.
Table 2
Specify the number of test environments and hybrids involved in this analysis.
Provide variable names for the columns.
Define all acronyms in the footnote e.g. BLUPs, CAN
Figure 1
Provide legends for blue, green and red items in the Figure.
Define G-I, -II and -III, and CAN
Table 3
Specify the number of test environments and hybrids involved in this analysis.
Edit the title as follows
Table 3. Minimum (Min), maximum (Max) and mean carotenoid concentrations I (CAN1) and II (CAN2) scores for three groups of hybrids defined based on canonical discriminant analyses.
How do we know CAN1 and II CAN2 in the Table.
Figure 2
Provide legends for blue, green and red items in the Figure.
What are the factors FA1 and FA2 stand for? Are these principal components – PC1 vs PC2? What were the loading scores in %?
Table 4
Specify the number of test environments and hybrids involved in the analysis.
Provide variable names for the columns.
Table 5
Specify the number of test environments and hybrids involved in the analysis.
Provide the variable names for the first two columns.
Define the acronyms in the footnote, e.g., FA1, FA2
Figure 3
Provide legends for blue, green and red items in the Figure.
What are the factors FA1 and FA2 stand for? Are these principal components – PC1 and PC2? What were the loading scores in %?
Supplementary Table 1.
Please add and label one column and mark/show the followings: experimental single-crosses, three-way crosses, commercial single cross and an orange endosperm commercial single-cross, and local variety. These can not be seen as it is.
Conclusions:
Overall, which of the hybrids were recommended providing higher grain yields, higher provitamin A and for being stable performers? For which country/region?
References
Authors have used a long list (83) of references. As a research paper the list should be substantially reduced by removing redundant and unrelated sources and using only high impact or good journals.
Author Response
//
Reviewer 2.
- What types of hybrids were the provitamin A biofortified hybrids? Single cross, three-way or four-way hybrids. Please enlist and distinguish these in the Supplementary Table 1. - Done
What are the field environments? Are these field conditions of multiple sites or seasons? As described in the Materials and Methods, environments represent location by year combinations (Lines 481-487 and 524).
Provide some values/figures related to higher grain yields for the new hybrids (e.g. > xx t/ha) that accumulated more than 10 µg/g of provitamin A compared with the existing orange commercial maize hybrid hybrids (e.g. > xx t/ha). Yield values provided in the abstract, results and discussion sections (Lines 20-21 and 599-603)
Overall, which of the hybrids were recommended for providing higher grain yields, higher provitamin A and for being stable performers? Seven hybrids (H08, H17, H23, H24, H41, H53 and H55) are included in the trait correlation and conclusion sections (Lines 288-289 and 613).
The following conclusion is debatable:-
‘Our study showed that breeding for enhanced provitamin A content had no adverse effect on grain yield in biofortified hybrids.’ Grain yield is often negatively correlated with enhanced nutritional content. Hence, low yields in biofortified varieties are the main cause for low adoption rate. Perhaps authors should reposition their conclusion to refer to ‘the present hybrid populations and field environments’ rather than providing general conclusions. - This statement has been modified in the abstract and the conclusion to accommodate the recommendation (Lines 24 and 606).
Introduction
Authors have provided extensive review on carotenoids and their bio-synthesis and accumulation. This should be kept to a minimum since the focus is to elucidate agronomic performance of provitamin A enriched tropical maize hybrids under multiple field conditions. We have been selective in citing only pertinent literate that relate to carotenoids and their potential association with agronomic performance in hybrids. The list of references is not excessive considering the large volume of research work conducted on carotenoids and reported in the literature.
Authors should provide sufficient background on the following aspects:
historical perspectives on breeding and genetic analyses for provitamin A enrichment; progress on variety development, registration and release of provitamin A enriched tropical or temperate maize hybrids; bottlenecks on adoption rate of biofortified maize; - A brief summary of the breeding emphases and progress has been provided in the second paragraph of the introduction with due citation of published breeding articles that cover these aspects in detail (Lines 48-60).
influence of production environments on the expression of provitamin A, yield and yield related traits; -Provided in the fourth paragraph of the introduction section (Lines 92-96).
correlations between yield and yield related quantitative traits and provitamin A and other nutrient contents. Any findings that reported breeding for enhanced provitamin A or nutrient content and its effect on grain yield in hybrids or OPVs. – The reviews of relevant literate in these areas have been provided in the third paragraph of the introduction and the last paragraph of the discussion section (Lines 71-84 and 431-439).
Before the objective statement, authors should relate this work to their current breeding efforts and the need and the cases for assessing provitamin A enrichment and agronomic performance under multiple field environments in their hybrid breeding program for recommendation, hybrid breeding etc. - Provided in the last paragraph of the introduction section (Lines 92-111).
Results
It will be interesting if authors can provide the mean values and some statistical contrasts amongst the tested hybrids for grain yield and nutrient profiles. The mean values should be cascaded by the test countries or selected regions/agro-ecologies and years of evaluation. I think one can make several inferences from this data set. Please provide this in a Supplemental Table. - The regional trials were conducted in 23 environments for carotenoid accumulation and agronomic performance in 103 environments. The environments that represent year-location combinations were not sampled to be representative at the country or regional level. As indicated in the Materials and Methods section, the datasets used for this analysis were unbalanced (not all hybrids are grown in all environments). For this reason, we used a mixed model with a factor analytic structure because this approach is more parsimonious and increases the precision of estimating the breeding values. Therefore, the data sets used for such analysis were unbalanced and could not generate results for selected regions and years to be include in a Supplementary Table.
Table 1.
Please specify the number of test environments and hybrids involved in this analysis. - Done
Provide variable names for the columns. - Done
Table 2
Specify the number of test environments and hybrids involved in this analysis. - Done
Provide variable names for the columns. - Done
Define all acronyms in the footnote e.g. BLUPs, CAN- Done (Lines 168-170)
Figure 1
Provide legends for blue, green and red items in the Figure. - Done
Define G-I, -II and -III, and CAN- Done (Lines 174-177)
Table 3
Specify the number of test environments and hybrids involved in this analysis. - Done
Edit the title as follows
Table 3. Minimum (Min), maximum (Max) and mean carotenoid concentrations I (CAN1) and II (CAN2) scores for three groups of hybrids defined based on canonical discriminant analyses. - Done
How do we know CAN1 and CAN2 in the Table. Provided in Supplementary Table 2.
Figure 2
Provide legends for blue, green and red items in the Figure. - Done
What are the factors FA1 and FA2 stand for? - Described (Lines 189-190) Are these principal components – PC1 vs PC2? - No. What were the loading scores in %? - Loading scores are provided as numbers for each environment not as percentages.
Table 4
Specify the number of test environments and hybrids involved in the analysis. - Done
Provide variable names for the columns. - Done
Table 5
Specify the number of test environments and hybrids involved in the analysis. - Done
Provide the variable names for the first two columns. - Done
Define the acronyms in the footnote, e.g., FA1, FA2- Done
Figure 3
Provide legends for blue, green and red items in the Figure. - Done
What are the factors FA1 and FA2 stand for? - Described (Lines 189-190) Are these principal components – PC1 vs PC2? - No. What were the loading scores in %? - Loading scores are provided as numbers for each environment not as percentages.
Supplementary Table 1.
Please add and label one column and mark/show the followings: experimental single-crosses, three-way crosses, commercial single cross and an orange endosperm commercial single-cross, and local variety. These can not be seen as it is. – Done.
Conclusions:
Overall, which of the hybrids were recommended providing higher grain yields, higher provitamin A and for being stable performers? Seven hybrids (H08, H17, H23, H24, H41, H53 and H55) are included in the trait correlation and conclusion sections (Lines 288-289 and 613). For which country/region? The focus of the manuscript is not to identify stable hybrid for each country/region.
References
Authors have used a long list (83) of references. As a research paper the list should be substantially reduced by removing redundant and unrelated sources and using only high impact or good journals. Given the extensive nature of available publications on carotenoids, we included only relevant list of references to the research area presented in the manuscript.
Reviewer 3 Report
Paper "Unravelling the Effect of provitamin A Enrichment on Agronomic Performance of Tropical Maize Hybrids" is very interesting.
Minor corrections:
"Zea mays" should be italic.
Correlation coefficients 'r' and p-value 'p' should be italic.
Quality of Figure 4 is poor.
Statistical methods used in this paper are corrected.
A combined analysis is correct for experiment in the following years.
In my opinion paper needs canonical variate analysis and estimation of Mahalanobis distance, but it is additionally analysis.
Paper needs minor revision.
Round 2
Reviewer 2 Report
The authors have done commendable job and provided considerable changes in the Tables, Figures and Supplemental Tables. These are clearly visible and enhanced the content.
However, I was unable able to assess the changes in the Introduction section per my earlier suggestion. Authors should have shown the new changes by highlighting the text or through track changes. I do not see any addition or reduction of references at this section or other sections of the manuscript. The number of citations still stand at 83 to warrant content change in synchronization with the relevant review of the literature.
Author Response
Please find below a point by point response to the second reviewer;
Authors have provided extensive review on carotenoids and their bio-synthesis and accumulation. This should be kept to a minimum since the focus is to elucidate agronomic performance of provitamin A enriched tropical maize hybrids under multiple field conditions. We have been selective in citing only pertinent literate that relate to carotenoids and their potential association with agronomic performance in hybrids. The list of references is not excessive considering the large volume of research work conducted on carotenoids and reported in the literature.
Authors should provide sufficient background on the following aspects:
historical perspectives on breeding and genetic analyses for provitamin A enrichment; progress on variety development, registration and release of provitamin A enriched tropical or temperate maize hybrids; bottlenecks on adoption rate of biofortified maize; - A brief summary of the breeding emphases and progress has been provided in the second paragraph of the introduction with due citation of published breeding articles that cover these aspects in detail (Lines 48-60).
influence of production environments on the expression of provitamin A, yield and yield related traits; -Provided in the fourth paragraph of the introduction section (Lines 92-96).
correlations between yield and yield related quantitative traits and provitamin A and other nutrient contents. Any findings that reported breeding for enhanced provitamin A or nutrient content and its effect on grain yield in hybrids or OPVs. – The reviews of relevant literate in these areas have been provided in the third paragraph of the introduction and the last paragraph of the discussion section (Lines 71-84 and 431-439).
Before the objective statement, authors should relate this work to their current breeding efforts and the need and the cases for assessing provitamin A enrichment and agronomic performance under multiple field environments in their hybrid breeding program for recommendation, hybrid breeding etc. - Provided in the last paragraph of the introduction section (Lines 92-111).